# Leveraging Remote Sensing Data for Yield Prediction with Deep Transfer Learning

**DOI:** 10.3390/s24030770

**Published:** 2024-01-24

**Authors:** Florian Huber, Alvin Inderka, Volker Steinhage

**Affiliations:** Department of Computer Science IV, University of Bonn, 53121 Bonn, Germany; s6alinde@uni-bonn.de

**Keywords:** remote sensing, yield prediction, deep learning, transfer learning, regularization, Gaussian process

## Abstract

Remote sensing data represent one of the most important sources for automized yield prediction. High temporal and spatial resolution, historical record availability, reliability, and low cost are key factors in predicting yields around the world. Yield prediction as a machine learning task is challenging, as reliable ground truth data are difficult to obtain, especially since new data points can only be acquired once a year during harvest. Factors that influence annual yields are plentiful, and data acquisition can be expensive, as crop-related data often need to be captured by experts or specialized sensors. A solution to both problems can be provided by deep transfer learning based on remote sensing data. Satellite images are free of charge, and transfer learning allows recognition of yield-related patterns within countries where data are plentiful and transfers the knowledge to other domains, thus limiting the number of ground truth observations needed. Within this study, we examine the use of transfer learning for yield prediction, where the data preprocessing towards histograms is unique. We present a deep transfer learning framework for yield prediction and demonstrate its successful application to transfer knowledge gained from US soybean yield prediction to soybean yield prediction within Argentina. We perform a temporal alignment of the two domains and improve transfer learning by applying several transfer learning techniques, such as $L^{2}$-SP, BSS, and layer freezing, to overcome catastrophic forgetting and negative transfer problems. Lastly, we exploit spatio-temporal patterns within the data by applying a Gaussian process. We are able to improve the performance of soybean yield prediction in Argentina by a total of 19% in terms of RMSE and 39% in terms of $R^{2}$ compared to predictions without transfer learning and Gaussian processes. This proof of concept for advanced transfer learning techniques for yield prediction and remote sensing data in the form of histograms can enable successful yield prediction, especially in emerging and developing countries, where reliable data are usually limited.

## 1. Introduction

Yield prediction is a challenging machine learning task. The relations determining the yearly yields are complex and depend on a variety of different factors. To predict yields, we need to measure plant condition and climatic states over an extended period of time. This monitoring is usually very costly and is dependent on domain experts or expensive sensors. A cost-effective solution is provided by using remote sensing data. Utilizing satellite images provided by huge organizations like NASA is free of cost and gives an extensive coverage of the phenological conditions of plants and the climatic state nearly worldwide. The second difficulty for yield prediction is the availability of historic ground truth yield data for training machine learning models. New data points can only be acquired once a year, so to train yield prediction models, we are reliant on historical databases that capture former yield quantities. Although databases like this are often available in developed countries, data are increasingly hard to come by for others. The aim of this study is to remove the gap in performance for yield prediction between countries with many data available and countries with few data available, especially since countries with fewer options to capture data are often also lacking other resources and could therefore greatly benefit from increased planning capabilities, which naturally arise from better yield prediction. As a solution to this problem, we leverage the concept of transfer learning to limit the number of ground truth yields needed for accurate predictions. The main idea is to use deep learning to adapt to the underlying relations that determine the yearly yields from remote sensing images and ground truth data in countries where data are plentiful, and to transfer this knowledge to yield prediction in countries with limited data. The difficulty of knowledge transfer for deep learning is determined by how close the two domains chosen to transfer knowledge are to each other. In this study, we explore the transfer of knowledge from the prediction of soybean yields in the US to the prediction of soybean yields in Argentina. The main difference between the two domains comes from the location of the two countries in different hemispheres of the globe and the two countries showing different climatic conditions and growth patterns for soy fields. We prepare the two datasets for transfer learning by performing a spatial-temporal alignment. Additionally, we adapt to common transfer learning problems such as catastrophic forgetting and negative transfer by applying the state-of-the-art regularization methods $L^{2}$-SP and Batch Spectral Shrinking (BSS). Furthermore, we find that a Gaussian process attached to our convolutional neural network (CNN) improves the accuracy of the yield prediction even further.

### 1.1. Related Work

The related work that influences our research can be categorized into three main groups. First, the general developments in transfer learning, as it is mostly used for computer vision tasks. Second, the advances of machine learning for yield prediction. And third, the advances of transfer learning for remote sensing applications.

#### 1.1.1. General Transfer Learning

Transfer learning is a topic of increasing popularity in different research areas. Most commonly, computer vision tasks are solved with the help of deep learning networks that are pretrained on huge task-agnostic datasets, before being fine-tuned to solve specific problems. Plested and Gedeon [1] give an in-depth survey of the state of transfer learning. Due to the versatile applicability of CNNs, the application of transfer learning methods to CNNs represents a well-known study objective. In image classification, transfer learning methods achieve significant success [2]. In this work, we examine whether this success can be translated into yield prediction while we follow the insights of the community. Huh et al. [3] found out that a larger dataset for pretraining results in a better model performance. Furthermore, the number of layers that should be transferred during the training process depends on the similarity between the target and the source dataset [4].

A common problem in transfer learning is the negative transfer of knowledge [5]. Negative transfer occurs when the source dataset used for pretraining and the target dataset are not well related, and transfer learning has a negative impact on the model accuracy. According to Plested and Gedeon [1], greater similarity between the domains of transfer learning improves the ability to transfer knowledge between domains. If the domains are not well aligned, the key to overcoming negative transfer is regularization which restricts the amount of knowledge that can be lost during the fine-tuning step of transfer learning. The $L^{2}$-SP regularization method achieves success in dealing with negative transfer through an $L^{2}$ regularization with the origin parameters of the more general model [6]. In our use case, this means that features that are extracted from the patterns of remote sensing data can be preserved during the transfer learning. Similar results can be achieved with DELTA regularization [7], following the idea of only altering CNN channels that are not already useful for the target task. Batch Spectral Shrinkage (BSS) is another regularization method that often successfully eliminates negative transfer by suppressing non-transferable spectral components [8]. Chen et al. [8] report that BSS will never negatively affect performance in a given dataset and it is therefore also considered within our work to stabilize transfer learning.

Our work is the first to examine the extended use of regularization and transfer learning on hyperspectral remote sensing data presented as histograms, investigating how ideas developed for classical computer vision can help us with remote sensing applications.

#### 1.1.2. Remote Sensing in Yield Prediction

More and more studies are addressing the investigation of yield prediction using machine learning [9,10]. In traditional machine learning, random forests, neural networks, and gradient boosting trees represent the most common methods of yield prediction [10]. Deep learning methods, especially CNNs and LSTMs, provide better feature extraction and generally lead to higher performance [9]. The basis for the data provided is mainly satellite imagery, which in many cases is enhanced with additional features [9]. You et al. [11] propose a novel and widely adopted dimensionality reduction technique to prepare irregularly shaped hyperspectral remote sensing images for regression tasks with deep learning. Remote sensing data from the Moderate-Resolution Imaging Spectroradiometers (MODIS) installed upon the NASA Terra and Aqua satellites are processed toward histograms, and both a CNN and an LSTM network are combined with a deep Gaussian process to perform a soybean crop forecast. The success of the deep Gaussian process built on an LSTM is confirmed by Kaneko et al. [12] for even sparser datasets, again using MODIS observations as input. Moreover, an LSTM trained in several countries without a Gaussian process results in a similar performance in predicting the districts of individual countries as the same model trained only in the country of prediction [12], showing the capabilities of the Gaussian process. Wang et al. [13] use, again, the same MODIS observations and data preprocessing to train an LSTM network to perform simple transfer learning in the form of pre-initialization of weights by a model trained on Argentina to predict soybean yields in Brazil. Similarly, Khaki et al. [14] use a CNN with a common backbone to simultaneously predict corn and soybean yields in the US. Regarding gradient boosting trees, the work of Huber et al. [15] shows on-par performance when compared with deep learning methods to predict soybean yield in the US based on MODIS data, while highlighting the improved explainability of gradient boosting trees and the reduced resources needed to train models. Gradient boosting trees is used along with a comparison including deep learning by Desloires et al. [16] to predict corn yield in the US based on Sentinel-2 satellite images.

The related work clearly suggests to us to also present MODIS observations in the form of histograms to our deep learning architecture. The data sources we consider for our study are also in line with other research in the field, including multiple wavelengths of surface reflections and surface temperature data from the MODIS satellites [10,11,13]. However, none of the previous works aimed to transfer knowledge between countries in different hemispheres of the globe. As different climatic conditions are challenging, this allows us to use the most extensive source of historic yield data in the US [17] to help with yield prediction all around the world.

#### 1.1.3. Transfer Learning for Remote Sensing Applications

Transfer learning for remote sensing applications is a widely investigated topic. A recent extensive survey on transfer learning on remote sensing data is done by Ma et al. [18]. Most commonly, the task of land use/land cover classification is improved by starting from pretrained models. Dastour and Hassan [19] give an in-depth analysis of different deep learning architectures for this task. In general, all CNNs tested are pretrained on the well-known ImageNet dataset [20] and applied to land cover classification on Sentinel-2A images. ResNet50 [21] was found to be the best architecture for transfer learning in this scenario. Li et al. [22] show that a ResNet architecture again works best on the HSRRS dataset for scene classification in urban built-up areas after being pretrained on the ImageNet dataset. Similar results are shown in the work of Alem and Kumar [23] where a ResNet50 architecture pretrained on ImageNet data shows high accuracy for land cover classification on remote sensing data. Tseng et al. [24] use the pretrained ResNet architecture inside Faster R-CNN as a backbone for rice seedling detection in RGB images outperforming a support vector machine. Chen et al. [25] use the same pretrained architecture, the Faster R-CNN network. First, features are learned on the ImageNet dataset before the network is fine-tuned to detect objects in high-resolution satellite images. Hilal et al. [26] extend the idea of a pretrained ResNet50 for land cover classification by including discrete local binary patterns to the ResNet features for the final classification. In very recent advances, Ma et al. [27] improve the idea of Domain-Adversial Neural Networks (DANNs) [28], where transfer learning is extended by projecting the input features of each domain into a common subspace. Their proposed partial DANN (PDANN) applies weights to the source samples according to their estimated yield distribution in the target domain and is used to improve the transfer learning of soybean and corn yields between different regions in the US.

Our work is quite distinguishable from the approaches described above. As our input data are processed to be represented by histograms and include multiple hyperspectral image channels and our target is not a classification but a regression, it is not possible to use huge image datasets like ImageNet as a source domain for transfer learning. Even in our source domain, we have relatively small numbers of training data when compared with modern image classification tasks. We need to overcome this issue by using deliberate regularization methods during knowledge transfer.

## 2. Materials

In this Section, we describe the two sources of data used within the study. We obtained ground truth yields for both countries from the relevant agricultural departments and used remote sensing data provided by the NASA Moderate-Resolution Imaging Spectoradiometer (MODIS) satellites for prediction.

### 2.1. Yield Data

Soybean yield data are taken from the US Department of Agriculture (USDA) for the US during the period 2010 to 2020, inclusive [17]. In Argentina, yield data are taken from the Ministerio de Agricultura [29] over the 2010/11 to 2020/2021 seasons, inclusive. This leaves us with a total of 8465 data points in the US and 1542 data points in Argentina. An overview of the yield data can be seen in Figure 1. Generally, we see that the yield data in the US and Argentina show similar patterns, with the relative yields in Argentina consistently lower than in the US. Both datasets show a positive linear trend that indicates higher soybean yields in recent history. This is caused mainly by technological innovations that lead to more efficient agriculture [30].

As described in Section 1.1.1, greater similarity between the domains of transfer learning improves the ability to transfer knowledge between domains. Here, the data for the US denote the source domain, and the data for Argentina denote the target domain. In order to keep the similarity of the data as high as possible, the same data cleansing procedure is used for both domains. In our case, data cleansing serves to ensure a sufficiently large input dataset. First, we select the regions of each country where soybeans are grown. This is the northeastern part of the US and the north of Argentina. Then, on the one hand, data points that cannot be assigned to a county due to missing information and counties with no yield specified or with a yield per hectare of zero are removed. On the other hand, all counties that have fewer than 2000 pixels of crop mask are removed. This is necessary to ensure a significant amount of soybean cropland for every data point and to reduce the number of noise from other sources. Figure 2 shows the counties used and discarded as a result in northeast Argentina and the eastern part of the US, leaving us with the main agricultural producers.

### 2.2. Remote Sensing Data

The use of satellite imagery for yield prediction is one of the main sources of reliable and cost-effective yield prediction. The MODIS satellite images we used were acquired on NASA’s Terra and Aqua satellites and are listed in the Google Earth Engine catalog. We used three different types of remote sensing data.

The surface reflections of light are captured in a range starting from the visible spectrum (459–479 nm) to the invisible infrared spectrum (2105–2155 nm) and are captured by 7 different channels. Each channel is referred to as a band in the context of remote sensing images. Specifically, we used version 6.1 of the MOD09A1 dataset [31] for surface reflectance corrected for atmospheric conditions. The final product is an 8-day composite of the images taken by the satellites with a resolution of 500 m by 500 m.

To monitor climatic conditions, we use a remote sensing dataset consisting of measured surface temperatures during day- and nighttime. The dataset used is version 6.1 of the MYD11A2 dataset [32]. The original resolution of 1 km by 1 km is resampled to match the resolution of 500 m by 500 m of the surface reflection data.

A cropland mask is used to filter out pixels that are not related to crop growths. Version 6 of the MCD12Q1 dataset [33] offers a yearly classification of several types of land cover with a resolution of 500 m by 500 m. Areas consisting of more than 60% cultivated fields are classified as croplands and will be the focus of our study.

Lastly, we crop the satellite images for the individual counties. Therefore, we require district assignments labeled with the county names. In the United States, the TIGER dataset from the United States Census Bureau [34] is used for this purpose. In Argentina, we use the county (Departamentos in Argentina) mapping used by Wang et al. [13], which contains all counties relevant to soybean cultivation.

## 3. Methods

We use multiple different tools to achieve the best accuracy for yield prediction on a smaller dataset. To allow for successful transfer learning, we align our two data sources first, before we apply multiple regularization techniques, and subsequently append a Gaussian process to recover the information lost due to histogramization. An overview of the process is shown in Figure 3.

### 3.1. Spatio-Temporal Alignment

The geographical distance between the US and Argentina accounts for some differences in soybean cultivation in both countries. Specifically, we need to adjust for different crop growth cycles. In the US, we monitor the growing conditions from March 23 to December 4. In Argentina, due to different climatic conditions, we monitor the crop growth cycle from November 26 to August 9. Both periods extend beyond harvest time, so that we can leverage the capabilities of our models to determine important input data themselves. As soybean yield prediction is especially valuable during the early stages of soybean growth, we examine a second in-season forecast to test our models. For this case, the time period only contains the first 14 of the 8-day intervals. The time periods used, as well as the underlying crop calendars [35], can be seen in Figure 4.

After data cleansing, a histogramization is performed, as was first done by You et al. [11]. For each band and each satellite image, the pixels masked as cropland within a county are summed for each record. Subsequently, these sums are transformed into a normalized histogram with 32 bins of the frequency distributions of the records in the vertical and the associated time in the horizontal. The resulting data point consists of 9 times 34 histograms, a county name, and a season year.

### 3.2. Model Design and Transfer Learning Techniques

At the center of our predictive model is a CNN, as was first used for yield prediction by You et al. [11]. As a baseline for our implementation, we started from the work of Tseng [36]. The CNN consists of 6 convolutional layers and the following dense layer. As a result of histogramization of the input data, we need an alternative solution to the usually used pooling layers in conjunction with the convolutional layers. The solution is convolutional layers with a stride of 2 to prevent locational invariance due to pooling. In addition, we use dropout layers, early stopping, and batch normalization when training the CNN. Primarily, to apply transfer learning, this CNN is trained in the US domain and will be referred to as the US model in the following. The most natural way to apply transfer learning is to fine-tune the US model in the Argentine domain. This corresponds to initializing the weights of the CNN for the Argentine domain with the weights of the US model. By assuming a similar distribution of the US domain and the Argentine domain, the starting point of the Argentine model will thus be chosen closer to the desired model, and the hypothesis space will be constrained by limiting the hyperparameters of the training. Furthermore, since it is known through extensive research that later layers have greater domain-specific significance while early layers extract general properties, it may be useful to distinguish the approach based on layer depth [1]. The first way to treat layers differently is to freeze some of them. The frozen layers keep their weights unchanged during fine-tuning and are effectively not trained directly on the new domain. In our case, this means that the frozen layers are trained only on the larger US domain to contribute to a prediction on the Argentine domain. Due to the high generalization of the front layers, this procedure is applied to the front convolutional layers and thus adopts the basic framework of the US model for feature extraction. This serves, in particular, to avoid catastrophic forgetting [37]. Catastrophic forgetting is one of the two major problems in transfer learning and is the tendency of neural networks to abruptly lose the knowledge of previous tasks when learning a new task [38]. Later convolutional layers are initialized with the weights of the US model but retain their ability to adapt to the Argentine domain. To name this retraining on top of the source domain, we overload the term fine-tuning, which also refers to the complete retraining of the net on the Argentine domain on top of the US model. Last, we use the ability to completely retrain layers independent of the US model so that they can fully adapt to the Argentine domain. This complete retraining is used in the dense layers to account for their strong specialization to the target domain associated with their depth.

Transfer-specific regularization methods are suitable to further limit catastrophic forgetting and to avoid negative transfer. Regularization influences CNN training by an additional summand Ω to the loss. The widely known $L^{2}$ regularization, also known as weight decay, penalizes large weights in the CNN. In this process, as seen in Equation Equation 1, the vector *w* of weights of the CNN is normalized, squared, and parameterized by α to set the regularization strength:(1)$$\begin{array}{c} \Omega \left(w\right)=\frac{\alpha }{2}{∥w∥}_{2}^{2}. \end{array}$$

Based on this, Xuhong et al. [6] modifies the starting point of this regularization so that instead of a deviation of the weights from zero, a deviation of the weights from the US model is penalized. Thus, the hypothesis space is again restricted to the surroundings of the US model, making catastrophic forgetting less likely. Since for the deviation of the target model from the source model an equal structure of both networks is required, Xuhong et al. [6] extends the model with the option to regularize non-transferable and especially newly added parts of the model with the $L^{2}$ regularization. Equation (Equation 2) shows the corresponding formula, where α and β set the strengths of the regularization of the constant and newly added parts:(2)$$\begin{array}{c} \Omega \left(w\right)=\frac{\alpha }{2}∥w_{S}-w_{s}^{0}{∥}_{2}^{2}+\frac{\beta }{2}{∥w_{\bar{S}}∥}_{2}^{2}, \end{array}$$
where $w_{S}$ is a weight vector of the constant structures of the target model, $w_{s}^{0}$ is a weight vector of the constant structures of the source model, and $w_{\bar{S}}$ is the weight vector of the newly added structures of the target model. We exploit the application of $L^{2}$ regularization for greater adaptability of the dense layer, so that dense layers are effectively regularized with $L^{2}$ regularization even when $L^{2}$-SP regularization is applied. This means in particular that the vectors of the constant structures are equal to the convolutive layers and the newly added structures are equal to the complete reinitialization of the dense layers.

Negative transfer refers to a loss of performance due to the transfer of knowledge and occurs due to a lack of transferability of some features caused by differences in the domains [1,8]. To reduce negative transfer, we use the Batch Spectral Shrinkage (BSS) regularization method. Chen et al. [8] observed that large datasets lead to highly generalized models, while at the same time these models suppress small singular values of the extracted features. To exploit this behavior of generalized models, Chen et al. [8] used artificial suppression of small singular values. Specifically, they introduced the concept of relative angles between domains that can be used to measure the transferability of eigenvectors in the weight matrices. Subsequently, they found out that in later layers of the network only eigenvectors corresponding to relatively large eigenvalues produce small angles and hence are well suited for knowledge transfer. Here, the feature maps $f_{i}$ of the input $x_{i}$ are vectorized and aggregated into a feature matrix $F=[f_{1}\ldots f_{b}]$ per batch of size b. Subsequently, the singular value decomposition is applied to this feature matrix, and the *k* smallest singular values are used to suppress the associated poorly transferable eigenvectors. The BSS regularization is thus given by
$$\Omega \left(F\right)=L_{bss}\left(F\right)=\eta \sum_{i=1}^{k}\sigma_{-i}^{2},$$
where η as a hyperparameter specifies the strength of the regularization and *k* specifies the number of smallest singular values $\sigma_{-i}$ to be penalized. *k* is set to 1 according to Chen et al. [8] and the regularization is completely adjusted via η. The use of BSS and $L^{2}$-SP has the advantage that both regularization methods can be applied simultaneously [8].

### 3.3. Deep Gaussian Process

To extend the CNN to include spatio-temporal features, You et al. [11] propose the use of a Gaussian process on top of the CNN. For a data point *x*, the deep Gaussian process uses the input of the last dense layer of the CNN h(x) and a Gaussian process $f\left(x\right)∼\mathrm{GP}(0,k\left(x,x^{\prime }\right))$ to make a harvest prediction:$$y\left(x\right)=f\left(x\right)+h{\left(x\right)}^{T}\beta$$

$f\left(x\right)∼\mathrm{GP}(0,k\left(x,x^{\prime }\right))$ is distributed from a Gaussian process with zero mean and covariance defined by the squared exponential kernel
$$k\left(x,x^{\prime }\right)=\sigma^{2}\;\exp\left(\frac{∥g_{loc}-g_{loc}^{\prime }{∥}_{2}^{2}}{2r_{loc}^{2}}-\frac{∥g_{year}-g_{year^{\prime }}{∥}_{2}^{2}}{2r_{year}^{2}}\right)\sigma_{e}^{2}\delta_{g,g^{\prime }}$$
where $g_{loc}$ and $g_{year}$ indicate the spatial and temporal data of the associated data point *x* and $\delta_{g,g^{\prime }}$ is the Kronecker delta as noise factor over $g=(g_{loc},g_{year})$ parameterized by $\sigma_{e}$. σ, $\sigma_{e}$, $r_{loc}$, and $r_{year}$ are hyperparameters. In summary, f(x) distributed from a Gaussian process is a collection of random variables with a joint Gaussian distribution. The next summand for the prediction is given by a set of basis functions h(·), corresponding to the last layer, and a random variable β. β is distributed from a normal distribution with the weight vector of the last layer as mean and $\sigma_{b}I$ as variance with the hyperparameter $\sigma_{b}$.

The choice of β and h(x) results in a distribution around the CNN prediction. The second summand then results from the covariance of the training data with the test data formed over the RBF kernel multiplied by the differences of the prediction $h{\left(x\right)}^{T}\beta$ on the training data and the ground truths of the training data, which are weighted again with the help of their covariance. This forms $f\left(x\right)∼\mathrm{GP}(0,k\left(x,x^{\prime }\right))$, which reduces the error values depending on the local and temporal relationships of the underlying data. For a deeper insight into the calculations, we recommend a look at the implementation of Tseng [36] and the explanations of Williams and Rasmussen [39].

### 3.4. Performance Metrics and Hyperparameter Tuning

Two main metrics are used to quantify the performance of the prediction. We measure the overall error by applying the Root Mean Square Error (RMSE), calculated as follows:(3)$$\mathrm{RMSE}\left(\hat{y},y\right)=\sqrt{\mathrm{MSE}(\hat{y},y)}=\sqrt{\frac{1}{n}\sum_{i=1}^{n}{\left({\hat{y}}_{i}-y_{i}\right)}^{2}}.$$

Here, y represents the ground truth values and $\hat{y}$ the prediction values. Furthermore, we measure the variability in the target variable explained by the models with the coefficient of determination (error):(4)$$\mathrm{error}\left(\hat{y},y\right)=1-\frac{\sum_{i=1}^{n}{\left(y_{i}-{\hat{y}}_{i}\right)}^{2}}{\sum_{i=1}^{n}{\left(y_{i}-\bar{y}\right)}^{2}},$$
where $\bar{y}$ is the mean of the ground truth values y. All models are evaluated over four runs each to account for the random factors of the model.

The models are trained from season 2010, respectively, 2010/2011 until one season before the prediction. The hyperparameters of the model are determined using the Optuna hyperparameter optimization framework [40]. Within Optuna, the number of adjusted hyperparameters is kept low to save computational resources and is estimated using a tree-structured parzen estimator algorithm. The search space of the hyperparameters was determined using Plested and Gedeon [1]. The remaining hyperparameters, such as those of the Gaussian process, are taken from Tseng [36]. Due to the required US data for information-breach-free application of transfer learning, the hyperparameters of the US model are derived using the three seasons from 2012 to 2014, resulting in an average number of 2457 training data points during hyperparameter tuning. For the Argentine model and the transfer learning parameters shown in Table 1, the seasons 2015/16 to 2017/18 are used to estimate the hyperparameters, resulting in an average number of 776 training data points during this step. The test data contain the years from 2018/19 to 2020/21 and contain 1268 training data points on average for Argentina and 7008 data points on average for the US. For every seasonal forecast in Argentina, we first train the US model with data from 2010 up to the previous season. This ensures that we have no information breach regarding temporal dependencies caused by transfer learning but always have the same data available that would be accessible in a real-world scenario.

## 4. Results

To analyze the capabilities of transfer learning for yield prediction, we analyze the accuracy of yield prediction models built with different configurations of regularization techniques and use of the Gaussian process. First, we examine the models over the long period, including the full crop growth cycle, before conducting experiments on the performance of an in-season prediction.

### 4.1. Full Growth Cycle Prediction

We compare the application of different transfer learning methods on the long period covering the full crop growth cycle, with a summary of the results being shown in Table 2. Additionally, a full overview of the results broken down to the performance each year is displayed in Appendix A. To account for the influence of random parameters, all values are determined over four runs each. For the Argentine models, the application of the Gaussian process improves the accuracy in all cases by up to 10% in terms of RMSE and by up to 25% in terms of $R^{2}$. Transfer learning approaches with regularization improve the results further. First, we examine the application of an initialization with the US model for all six layers with a freezing of the parameters of the first four layers. We see initial improvements compared to a model without any transfer learning applied that diminish after the concatenation of the Gaussian process. Incorporating the regularization techniques explained in Section 3.2 stabilizes the training and subsequently gives the best results. The overall best configuration is an initialization with the weights of the US model, a freezing of the first four layers, a fine-tuning using $L^{2}$-SP regularization, and lastly the application of the Gaussian process. Using BSS as an additional regularization slightly decreases the average accuracy but stabilizes the training process, as can be seen in Figure 5.

Figure 5 shows the distribution of the RMSE in bu/ac for each method. The first two boxplots show the basic drop in RMSE when the Gaussian process is applied. The US initialization with freezing subsequently causes a wide dispersion of the error values. While BSS places the center of these scattered error values at a low level, $L^{2}$-SP causes the values to be centered at a lower value, indicating a lower average error. The simultaneous use of BSS and $L^{2}$-SP further reduces the scattering of error values, indicating that the predictions provide greater reliability.

### 4.2. In Season Prediction

As described in Figure 4, we examine a second shorter time frame of available information for our prediction models. A model capable of inferring the estimated yield long before harvest has a very high value for crop management. Table 3 shows a summary of the prediction results in the short period. The full breakdown of the results for every individual year can be found in Appendix B. Here, the satellite images no longer include the harvest and end before the first harvests begin. Training, evaluation, and testing are performed exactly as in the long period. We used only the first 14 satellite images, instead of 32 as done previously. This is reflected in a reduced performance compared to the long period. At the same time, it can be seen that transfer learning methods, especially with simultaneous application of the Gaussian process, result in an even greater increase in performance.

## 5. Discussion

With our experiments, we are able to confirm the capabilities of CNNs for yield prediction when they are trained on MODIS observations that are preprocessed to resemble histograms. The approach is tested within the literature for several yield prediction scenarios, as described in Section 1.1. The addition of the Gaussian process for predicting US yields is beneficial for the short prediction period, while it decreases accuracy for the long prediction period. For the Argentine yield prediction without transfer learning, the Gaussian process improves the results by ca. 10% in terms of RMSE for the long period and the short period. Similar results are discovered by Kaneko et al. [12] where a Gaussian process improves the results of predictions based on small datasets up to a level similar to that which can be observed when training with more data. The same effects are shown when examining the first step of transfer learning for Argentina, where the CNN weights are initialized with the US model’s weights and the first four layers are frozen. Although the improvement without a Gaussian process is about 7% in terms of RMSE, the results are worse when the Gaussian process is included. Obtaining worse results with a machine learning model after application of knowledge transfer, like we have in the case of our model when the Gaussian process is included, can be described as negative transfer and is well anticipated in the literature (Section 2.1). As yield prediction is a research area that suffers from data scarcity, the selection of a source domain for pretraining is prone to a known tradeoff. On the one hand, more data for pretraining improves the results, while on the other hand, less similarity between the domains endangers the knowledge transfer. Our experimental setup tends to emphasize the amount of training data for pretraining over the similarity of the domains, as in real-world yield prediction applications it is often not possible to produce more training data close to the target domain. Despite this choice, the first transfer experiments indicate that features extracted from remote sensing data can be transferred similarly to those obtained in many computer vision tasks. This claim is supported by the fact that we were able to freeze the first four layers in our CNN which commonly condense high-level features from the data and improve the model accuracy without the Gaussian process.

The negative transfer that is prevalent when we evaluate the models with the Gaussian process can be addressed by regularization techniques. Our results support the claim that regularization techniques designed for commonly used image features have similar effects on remote sensing data presented as histograms. The $L^{2}$-SP regularization together with the Gaussian process gives the best results in terms of average RMSE and $R^{2}$, removing the negative transfer that occurred without regularization. As indicated by Chen et al. [8], the inclusion of BSS gives us a small decrease in average performance by 2.8% in terms of RMSE but stabilizes the knowledge transfer. As can be seen in Figure 5, the worst error values are closer to the average RMSE than in any other constellation, which makes us recommend the combination of layer freezing, $L^{2}$-SP, and BSS for transfer learning tasks, including remote sensing data represented by histograms. The same constellation also works well when considering the short prediction period (Table 3).

Regarding the specific transfer of knowledge between the US and Argentina, we can analyze Figure 6, providing an overview of the distribution of errors in Argentina after applying different procedures for the long prediction period. The graphics shown are taken from the first run of 2019 as an example. The area shown includes soybean acres in northeastern Argentina. Training or test data are missing for the gray areas. On the basis of the lower saturation of the districts, it can be seen that the Gaussian process reduces the error values. The application of transfer learning leads to low saturation in many districts in advance and a more balanced distribution of overestimates and underestimates. At the same time, especially in transfer learning, some districts with a worse forecast are visible. We need to assume that the relations learned in the US domain do not carry over to these districts, due to local dependencies or different crop management techniques. All in all, we find the regularized transfer learning to be beneficial and generally reduce the error on our datasets.

Within the wider literature context, the first parallel between our work and related work also considering yield prediction with remote sensing data is the use of MODIS satellite data as a primary data source, as is done, for example, by [11,13,14]. While all those works report good results, it is worth mentioning that alternative data sources exist. The authors of [41] also use the Sentinel-2 satellite to achieve state-of-the-art results, exploiting the fact that the images are available at a higher resolution. Furthermore, the literature is beginning to investigate the use of non-fixed time steps during histogram creation as input for machine learning procedures [16], which may be useful to increase alignment between the two domains used for transfer learning in the future. When considering the literature for yield prediction in general, our results improve on state-of-the-art performance, as we can deduce from our comparison to successfully deployed deep learning architectures [11] trained and evaluated in our data. This increase in performance comes from enabling the US, as the biggest repository for ground truth yield data, as the source domain for transfer learning, even for a country in a different hemisphere. This builds on the work of [13], where transfer learning is first used to improve yield prediction via transfer learning, but the two countries used, i.e., Argentina as source and Brazil as target, are much more closely related than in our case.

However, it is important to mention that the improvements come at the price of many additional hyperparameters that have to be tuned. Tuning hyperparameters in a deep learning context is always difficult, since the impact of a hyperparameter can mostly only be observed after a significant number of computations. This makes it so that research often turns to empirical values or educated guesses. Regularization and transfer learning include an additional six important hyperparameters to adjust: $L^{2}$-SP, BSS, the number of frozen layers, and the initialization of the non-frozen layers with parameters of the source model, as can be seen in Table 1. In our experiments, the number of four frozen layers indicates that many features learned from remote sensing data in our target domain can be directly transferred to the new task. The high transferability is also indicated by the advantageous initialization with the source weights for the non-frozen layers. The $L^{2}$-SP hyperparameter α with a value of 0.23 quantifies the punishment for altering the weights of the source models. This value being relatively low means that the non-frozen layers must be able to be highly adjustable to the new task, hinting that the yield-related patterns utilizing the frozen features of the first layers are quite different for both our yield prediction tasks. The high $L^{2}$-SP hyperparameter β is the standard $L^{2}$ punishment for high weights suppressing overfitting. The BSS hyperparameters η and *k* and their respective values 0.07 and 1 indicate that the strength of the regularization is relatively low and the smallest singular value is penalized. For all these hyperparameters, small adjustments can alter the models’ performance, increasing the risk of a bad model due to careless handling of the hyperparameters compared to a simpler model without transfer learning.

## 6. Conclusions

Overall, our proof of concept shows that transfer learning can lead to improved crop prediction using CNNs, in particular through a joint application of several methods with careful determination of the hyperparameters. In addition to the usual procedures of initializing the weights by a model trained on a larger dataset and fixing these weights, transfer-specific regularization methods with simultaneous application of the Gaussian process particularly lead to improved prediction. For prediction over the full crop growth cycle, performance was improved by an RMSE of 0.51 bu/ac compared to the CNN with Gaussian process and without transfer learning by using a US initialization of the weights, fixing four convolutional layers, and applying $L^{2}$-SP. The same procedure results in a performance increase of 1.35 bu/ac for the in-season prediction. The BSS regularization method is not directly beneficial to model performance, as is the case for $L^{2}$-SP, but it helps to stabilize the training process and prevents a particular bad prediction. We see that we can utilize transfer learning methods designed for other deep learning tasks, such as image classification, for yield prediction with remote sensing data in the form of histograms as input. Future work includes investigations in other countries, especially with even smaller yield databases, to further confirm these methods. In general, our approach for leveraging remote sensing data for yield prediction with deep transfer learning shows that:Spatio-temporal alignment can be performed even between two varying remote sensing data sources to allow for transfer learning.The capabilities of transfer-specific regularization methods $L^{2}$-SP and BSS together with Gaussian processes for transfer learning translate to the context of yield prediction and hyperspectral remote sensing data in the form of histograms.Regularized transfer learning can improve yield predictions in regions where fewer data are available and should be considered as an alternative to state-of-the-art approaches, especially for smaller study areas.

## Figures and Tables

**Figure 1 sensors-24-00770-f001:**
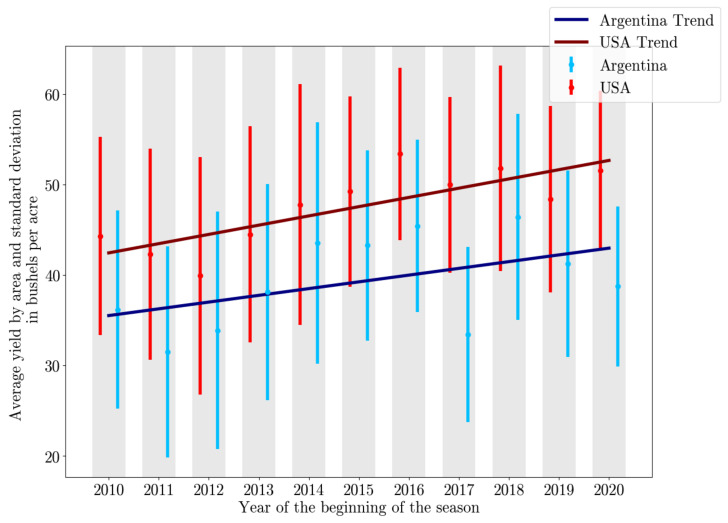
Visualization of the ground truth yield data in the US (red) and Argentina (blue). Average yields are indicated by the points and standard deviation is shown by the relevant lines, both in bushels per acre. The dark red and blue lines indicate the linear trend in the data.

**Figure 2 sensors-24-00770-f002:**
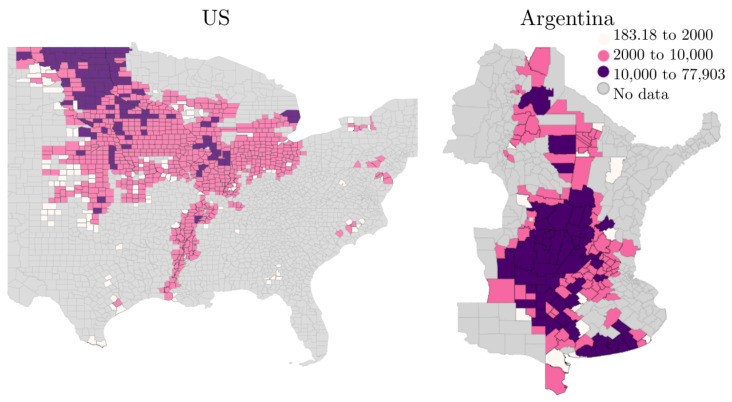
Average number of cropland pixels over the 11 years from 2010 to 2020 in the eastern part of the US and the northeast of Argentina. The pink to purple areas show the counties used. No cropland pixels or yield data were found in the gray areas, and too few in the white areas.

**Figure 3 sensors-24-00770-f003:**
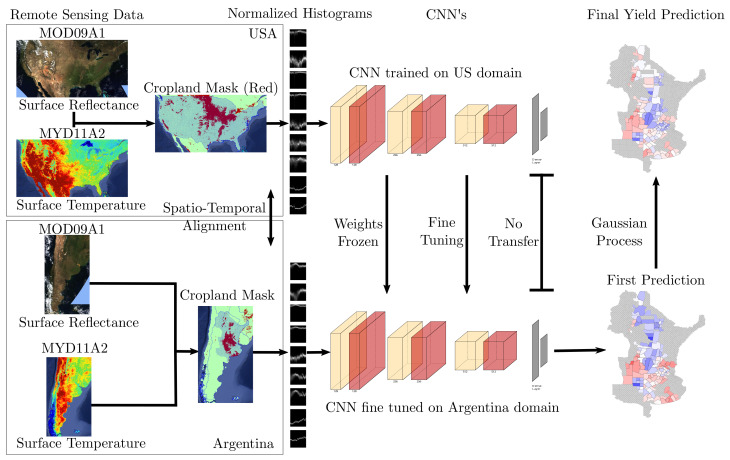
Overview of our approach for leveraging remote sensing data for yield prediction with deep transfer learning. The two boxes on the left side show the kind of input data that we use, consisting of surface reflectance data, surface temperature data, and the cropland mask for each of the two domains (Section 2). The data is spatio-temporally aligned as preparation for transfer learning (Section 3.1), before being processed toward normalized histograms (Section 3.2). The middle section of the figure shows some of the transfer learning techniques used: frozen weights, fine-tuning with regularization, and the initialization of the dense layer at the end of the network (Section 3.2). Lastly, on the right-hand side, we show an example of refining the prediction results by applying a Gaussian process (Section 3.3).

**Figure 4 sensors-24-00770-f004:**
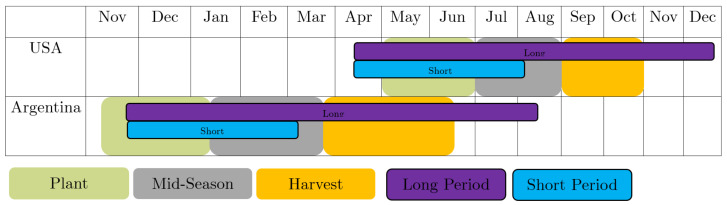
Crop calendar of Argentina and USA (data from USDA [35]). The data of the long period are shown in purple. The data of the short period are shown in blue.

**Figure 5 sensors-24-00770-f005:**
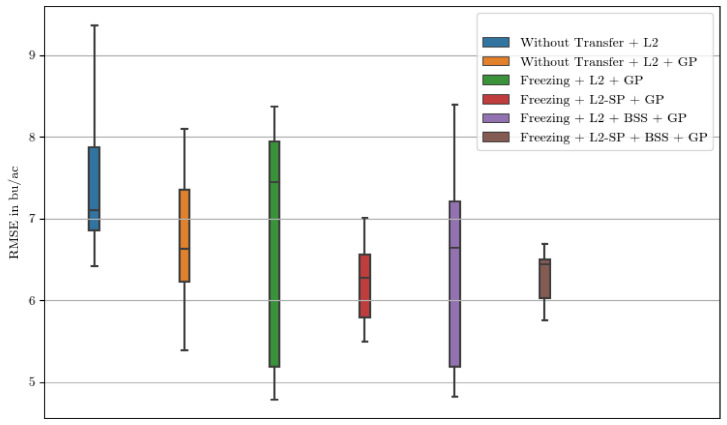
Distributions of the averaged district errors of all runs in all test years as boxplots. In this case, freezing implies the initialization of the 6 convolutional layers by the US model without further fine-tuning of the first 4 layers.

**Figure 6 sensors-24-00770-f006:**
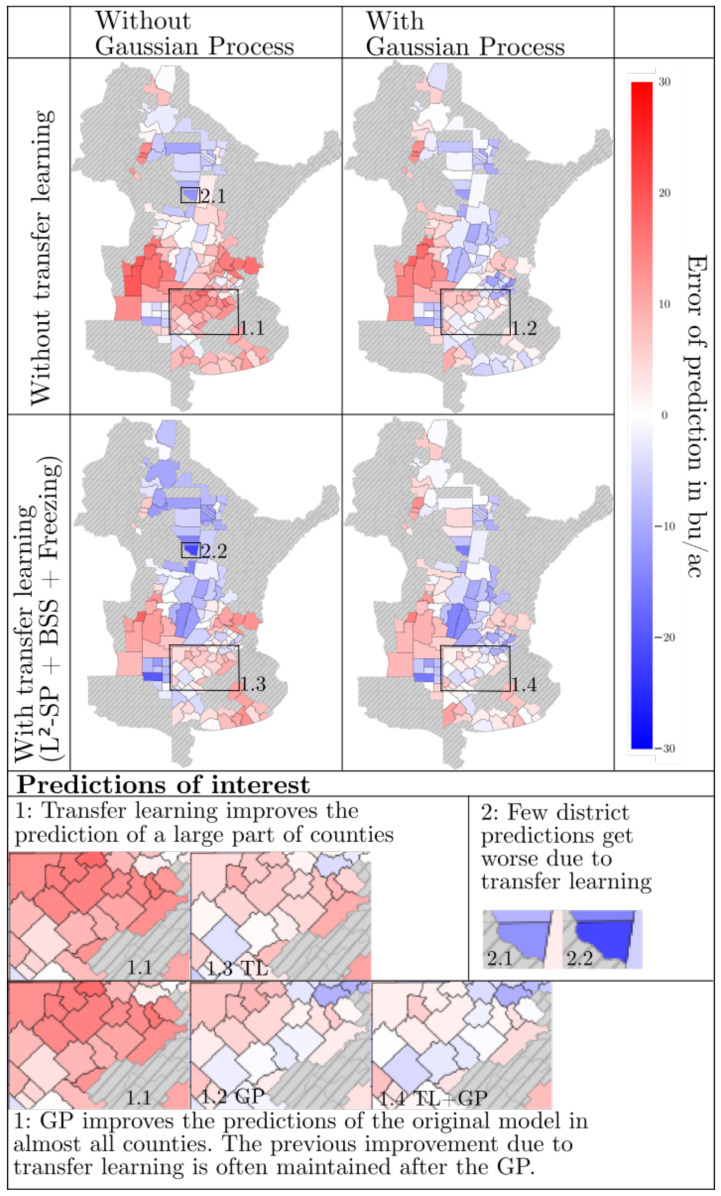
Comparison of different results in the first run of the long period of 2019. Districts without ground truth are grayed out. An overestimate is indicated in red and an underestimate is indicated in blue. The lower the saturation, the better the prediction.

**Table 1 sensors-24-00770-t001:** Hyperparameter configuration for transfer learning-related hyperparameter. Hyperparameters are either tuned via the tree parzen estimator (TPE) within the Optuna Framework or estimated empirically. The optimal hyperparameters are rounded and refer to the application of all methods in the long period, as shown in the last row of Table 2.

Hyperparameter	Tuning Method	Optimal Parameter
Number of frozen layer	Empirical	4 ∈ [0, 6]
US init. of weight	Empirical	True ∈ [True, False]
$L^{2}$-SP α	TPE	0.23 ∈[0,1]
$L^{2}$-SP β	TPE	0.77 ∈[0,1]
BSS η	TPE	0.07 ∈[0,1]
BSS *k*	Empirical	1 ∈[0,1]

**Table 2 sensors-24-00770-t002:** Average RMSE in bu/ac and $R^{2}$ as fraction of 1, of different model configurations for the long period. The long period contains satellite images beyond the harvest time and thus includes the full crop growth cycle. Bold numbers indicate the best scores. A full display of the results is shown in Appendix A.

	RMSE ↓	$R^{2}$ ↑	RMSE + GP ↓	$R^{2}$ + GP ↑
USA: CNN	5.94	0.583	6.81	0.439
Argentina without transfer	7.47	0.442	6.76	0.554
Argentina + freezing	6.94	0.526	6.85	0.547
Argentina + freezing and $L^{2}$-SP	6.80	0.550	**6.25**	**0.618**
Argentina + freezing, $L^{2}$ and BSS	7.05	0.516	6.43	0.593
Argentina + freezing, $L^{2}$-SP and BSS	7.07	0.511	6.31	0.608

**Table 3 sensors-24-00770-t003:** Average RMSE in bu/ac and $R^{2}$ as fraction of 1, of different model configurations for the short time period. Satellite image coverage ends before the start of the harvests. Bold numbers indicate the best scores. A full display of the results is shown in Appendix B.

	RMSE ↓	$R^{2}$ ↑	RMSE + GP ↓	$R^{2}$ + GP ↑
USA: CNN	7.12	0.394	7.00	0.414
Argentina without transfer	9.36	0.140	8.37	0.314
Argentina + freezing, $L^{2}$-SP and BSS	8.20	0.349	**6.92**	**0.537**

## Data Availability

All data used within this research are publicly available and cited accordingly.

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
