# Peer review of "Leveraging Remote Sensing Data for Yield Prediction with Deep Transfer Learning"

_sensors, 2024, doi:10.3390/s24030770_

Round 1
Reviewer 1 Report
Comments and Suggestions for Authors
Review of the manuscript entitled
Leveraging Remote Sensing Data for Yield Prediction with Deep Transfer Learning
The paper should be re-formatted according to the typical layout of a research paper, which includes chapters such as Introducton, Materials and Methods, Results, Discussion, Conclusion.
I have provided some guidelines below to assist Authors in preparing a revised version of the paper.
Title
Leveraging Remote Sensing Data for Yield Prediction with Deep Transfer Learning.
I would ask the Authors to consider whether the title should be made more specific, so it seems appropriate to include the USA and Argentina?
Abstract is well prepared.
Introduction
The initial section of the Introduction introduces the issue of yield prediction from the ground up. However, the later part is questionable.
Page 2 line 53-54 This sentence should be in the chapter Material and methods
Page 2 lines 55-62 are appropriate to be placed in the Summary or Conclusions, not in the Introduction
The objective of the research is not indicated in the Introduction.
Chapter 2 Related Work should be a continuation of the Introduction. Some parts of Chapter 3. Data description, which is essentially a literature review, should be in the Introduction.
I believe that the Authors should separate the chapter "Material and methods" in the paper. Many parts of Chapter 3. Data description describe the research data.
Methods seem to be adequately described
The paper lacks a "Results" chapter
Discussion
The discussion is rightly focused around the authors' own results, but many of the individual issues lack a wider literature context.
Conclusion
There is a lack of clear indication of the unique achievements created by the research.
Figure 1
The description under Figure 1, although appropriate Figure 1 is not compatible with it. I encourage the authors to make corrections, which I do not recommend in detail. It seems to me that the phrases: surface reflectance and surface temperature should be given under the maps of Argentina.
Reviewer 2 Report
Comments and Suggestions for Authors
Section 2: The literature review is not comprehensive. For example, Dr. Ma (https://scholar.google.com/citations?user=ZFqkBkgAAAAJ&hl=en) has done a lot of transfer learning work on yield prediction and a review on transfer learning in environmental remote sensing.
Line 176: What kinds of crop masks do you use?
Line 178: How many year-county samples are there in the final dataset?
Line 283-284: In Batch Spectral Shrinkage, the k smallest singular values are used to suppress the associated poorly transferable eigenvectors. It is unclear why those eigenvectors with low singular values have poor transferability.
Line 337: it is not clear why the authors use the long period covering the full crop growth. Since this study focuses on soybeans, it makes more sense to use the observations covering the growing and harvest seasons of soybeans instead.
How many samples in the target domain are used to fine-tune the USA model?
Line 383: It is not clear to me why “This behavior can be described as negative transfer”. Gaussian process embeds the spatial and temporal information based on the output of the last CNN dense layer. It is supposed to learn the domain-specific information. Also, why does the drop in performance only happen to the full crop growth dataset?
Table 1&2: Are the overall results in all testing years? It would be better to show the results in each testing year to check the consistency of the model performance.
Line 417: “We need to assume that the relations learned in the US domain do not carry over to these
districts, due to climatic conditions or different crop management techniques.” Do you have any specific examples/insights on what causes the larger domain differences in these districts compared to others? In Figure 6, the district on the east side of 2.2 has better accuracy under transfer learning. It indicates that the climate conditions are not the driving factors to the performance drop.
Round 2
Reviewer 1 Report
Comments and Suggestions for Authors
I accept the manuscript as revised by the authors. However, I regret the lack of response to my concerns regarding the layout of the content in the paper.